# Anomalies Detection and Proactive Defence of Routers Based on Multiple Information Learning [note 1]

**DOI:** 10.3390/e21080734

**Published:** 2019-07-26

**Authors:** Teng Li, Jianfeng Ma, Yulong Shen, Qingqi Pei

**Affiliations:** 1School of Cyber Engineering, Xidian University, Xi’an 710071, China; 2School of Computer Science, Xidian University, Xi’an 710071, China; 3Shaanxi Key Laboratory of BlockChain and Security Computing, Xidian University, Xi’an 710071, China

**Keywords:** router security, data correlation, attack detection

## Abstract

Routers are of great importance in the network that forward the data among the communication devices. If an attack attempts to intercept the information or make the network paralyzed, it can launch an attack towards the router and realize the suspicious goal. Therefore, protecting router security has great importance. However, router systems are notoriously difficult to understand or diagnose for their inaccessibility and heterogeneity. A common way of gaining access to the router system and detecting the anomaly behaviors is to inspect the router syslogs or monitor the packets of information flowing to the routers. These approaches just diagnose the routers from one aspect but do not correlate multiple logs. In this paper, we propose an approach to detect the anomalies and faults of the routers with multiple information learning. First, we do the offline learning to transform the benign or corrupted user actions into the syslogs. Then, we construct the log correlation among different events. During the detection phase, we calculate the distance between the event and the cluster to decide if it is an anomalous event and we use the attack chain to predict the potential threat. We applied our approach in a university network which contains Huawei, Cisco and Dlink routers for three months. We aligned our experiment with former work as a baseline for comparison. Our approach obtained 89.6% accuracy in detecting the attacks, which is 5.1% higher than the former work. The results show that our approach performs in limited time as well as memory usages and has high detection and low false positives.

## 1. Introduction

A router is a device that forwards data packets between computer networks. Routers are widely used in the Internet and have become the traffic center of the network flow. There are a lot of services, such as the web, file transfer, domain name resolution and dynamic host configuration server, running on these embedded devices. Even if they run as regular small computers and have the corresponding secure configuration choices, there is no anomaly detection of prediction software on them. They depend on firmware updates and firewalls to protect their security. Nowadays, routers are easily attacked by hackers. First, the users do not have professional knowledge about device security and set the device improperly. Second, the router is the data forwarding center of the network, and it is worthy for the hackers to attack them. Third, they are constantly online to forward data and router systems can have inherent bugs and misconfigurations installed by the developers or administrators. Besides, all network traffic passes through the routers. Thus, if an adversary manages to attack the routers, it can make the whole network paralyzed or gain the data control rights of the whole system. There are some recent attacks towards mainstream routers shown in Table 1.

Due to the vulnerability of the routers, many methods have attempted to diagnose routers. Labovitz et al. [1] analyzed the commercial Internet traffic to find the changes and provided the trend of the traffic of the network and the communication devices. Lee et al. [2] used the method based on MapReduce to do the flow analysis of the routers. However, not all of the router attacks can be reflected in traffic change. There are numerous works that concentrate on the routers’ syslogs. Routers record the events occurring on them by logging these messages, which contain debugging or error messages, internal states and exceptions. Some current anomaly detection based on log analysis are inappropriately used in routers. The authors of [3,4] needed to combine the source code of the systems to do the analysis. Although these works can improve the attack detection accuracy depending on source codes, they are not practical in real router systems. Apparently, most routers are closed systems where we cannot get the source codes in most cases, and reverse engineering of the firmware encounters many difficulties and obstacles [5,6]. Some researchers also concentrate on pure log diagnosis. Dlog [7] extracted the log templates and presented the cluster of the attacks. DeepLog [8] constructed the workflow from system logs to detect the anomalies through deep learning. Dusi et al. used network traffic such as the Tcpdum log [9] to monitor the network and find missing segments. Balzarotti et al. [10] constructed a system-call trace and analyzed system-call logs to detect if there were any malicious codes. However, all these works only use the syslogs as the single diagnosis source and ignore other information of the system, which is also significant in the anomaly detection. According to our survey of routers, relying solely on syslog messages cannot record all events and error messages on the device. Log information is recorded in the device at different granularities. If the system status enters a predetermined state, the corresponding log information will be printed. Therefore, the router log can form a limited state machine [11]. However, there are countless rampant attacks in the network and there are some modified versions for router attacks. Some attacks and anomalies can no longer be reflected in a single source log. Further, many attacks are no longer recorded in the router’s single log. Therefore, to detect these attacks, it is necessary to correlate and analyze multiple types of log information. Therefore, we need to use the correlation log information to improve the accuracy of the anomaly detection. Instead of using just one source log information as in works [8,12], we leverage multiple sources of logs, such as syslogs, firewall logs, CPU utilization, LAN status, and memory information to do the training step. With labeled data, we can learn the benign and malicious events. However, there are some events for which we cannot get labels before detection. Thus, we use the correlation analysis to perform clustering, which can help us migrate the labeled training problem. To accelerate the processing of a prohibitive volume of log data, data mining and machine learning are useful in solving this problem. Our approach consists of three steps: (1) learn the normal and abnormal states of the routers with the correlation of multiple log data; (2) detect anomalies with the input of multiple audit data combined with the result of the first step; and (3) use the longest common subsequences algorithm to find the regular pre-steps before the attack and use this attack chain to do the prediction in order to achieve proactive protection. We applied our approach on the real router network, and the results show that our approach can improve the current intrusion detection techniques. In summary, our key contributions are: (1) We use multi-source logs in the router for offline learning and training, which obviously improves the accuracy of attack detection. (2) We use correlation analysis method to get the relationships among events and find some unlabeled events during the learning step. (3) We perform anomaly detection by calculating the distance between the event and the clusters, and accurately classify the anomaly. (4) We use LCS algorithm to find the pre-steps before the attack, and we can use this chain to predict the attack before it happens.

## 2. Related Work

Log templates extraction: Qiu et al. [13] proposed the log template extraction method for router syslogs. They got the idea from the email-spam detection by using a signature sub-tree whose root is the same message type of the logs. The authors of [14] used a mixture of Hidden Markov Model (HMMs) from the original logs to model sequence patterns. However, these approaches only concentrate on normal numerical variables. Besides, they cannot extract various normal type logs. We acknowledge that logs also contain text parameters and our approach can also deal with these type of messages. STE [15] takes another method using the following features of log messages: parameters appear less frequently than template words. The approach gives a score to each word in the log and decides if the word belongs to the templates by using DBSCAN algorithm. The log messages can be massive and the word score may be inaccurate for repeated words in some formats. Kimura et al. [16] classified the words into five classes and gave each class a weight value indicating its template tendency. However, they neglected the position information of the word since a specific word normally has its unique position in one template. Log correlation analysis: Log correlation techniques [17,18,19] are widely used in the attack diagnosis field. They make causality analysis of DNS logs, HTTP logs, WFP logs and system logs of the communication platform or device. Then, they correlate these logs to achieve attack provenance. However, many existing logging techniques are too coarse-grained (e.g., Cisco routers record the unauthorized IP in both malicious events and benign events). When analyzing the attack chain and root causes, the coarse granularity will lead to the dependence explosion problem [20,21]. This makes it difficult to identify the real events related to the attack. Besides, many log correlation analysis approaches utilize parameter-based causality analysis [22,23]. This can also cause significant false relations because they do not uncover the real logical or semantic relations among logs or events [15]. Furthermore, it is difficult to define or identify the events in log entries [7]. Many approaches depend on slide time window to solve this problem [8]. However, setting the time window is ambiguous and the heavy workload makes it nearly impossible to be used in a real system. Anomaly detection: Using the syslogs to do the anomaly detection is widely researched in router device. The authors of [24,25] suppressed many less important and usual log messages to find hidden anomalies by using the global weight. Because only once appearing events have a high weight, the methods cannot confirm the clear difference of the results in anomaly detection. The diagnostic tool FDiag [26] identifies significant events that lead to compute node soft lockup failure by using the combination of message template extraction, statistical event correlation and episode construction. However, its diagnostic capabilities are limited to identifying the dates of event sequences and the correlated events for only one time period. It is successful at identifying the causal relationships but does not identify the nodes and it requires manual execution of the framework’s workflow functions. All of these works only focus on anomaly detection and do not consider the attack prediction and prevention.

## 3. Overview and Roadmap

The system consists of three main steps which is shown in Figure 1: (1) learn the normal and abnormal states of the routers with the correlation of multiple log data; (2) detect anomalies with the input of multiple audit data combined with the result of the first step; and (3) use longest common subsequences algorithm to find the regular pre-steps before the attack and use this attack chain to do the prediction in order to achieve proactive protection.

### 3.1. Offline Learning (Events → Logs)

In this phase, we try to find a clear relationship between the event and the router’s multi-source log. We map events to the corresponding log information characteristics. This approach reveals the behavior of different types of attacks. In the learning phase, we use the perspective of the administrator and control the router. Administrators can perform normal and malicious behavior on the routers. Furthermore, the corresponding multi-source log information can be observed and we can use these data to perform the following data analysis work. The advantage of this step is that we do not need to analyze all the data information of the router and we can know the relationship between user behavior and log information. However, there are also some unlabeled data that we use the log correlation analysis method to solve. We try to find the relationships among the events and put them in the same cluster according to the strength among their ties.

### 3.2. Anomaly Detection (Logs → Events)

In this step, we try to detect the anomaly and we consider the problem from the perspective of the user. At this point, we only get the log information. First, we collect the relevant router multi-source log information and try to identify the event in the log information. Next, we examine the corresponding information of this event and do the anomaly detection. Finally, we try to cluster the attack to a specific cluster and tell users which attack it is. In this phase, we try to translate the log to the corresponding behaviors and do the outlier detection. The advantage of this approach is that we can detect the attack pretty quickly according to the distance calculation. Data mining is useful in this approach in identifying the event and the cluster approach can find the information violation during this phase.

### 3.3. Anomaly Prediction (Logs → Events → Attack Chain)

Only detecting the anomalies is not enough for us. Because we intend to protect the routers security proactively, we must predict the attack before it happens. As we have spotted the attack, we trace back the logs and find the regular pre-steps before the attack. We regard each event before attack as an alarm of the whole attack chain and we use longest common subsequence algorithm to find the adversary’s sneaking actions. With the found attack chain, we can predict the anomalies by spotting the current condition or event on the routers. Thus, we can achieve the goal of protecting the routers in advance without the attack happening.

## 4. Methodology

Our primary goal is to prove that multi-source log information can improve the accuracy of abnormal event detection. To achieve this, we need to know the relationship between user behavior and multi-source logs and identify events and anomaly detection through multi-source log information. In this section, we explain how different user behaviors are recorded in different logs and how they can be used to detect different attacks.

Table 2 shows how a sampling of actions (Nos. 1–3 are normal user events and Nos. 4–6 are attacks) affects different logs. After analyzing how actions reflect on each log, we can identify some important logs that need to be checked primarily: Syslog, CPU utilization, LAN status and Memory utilization. We not only analyze the normal user events but also the attacks, which is different from the analysis approach in [27]. All of the logs can be obtained from the router log system as we login as administrator and this does not need any extra help from the developers.

### 4.1. Data Preprocessing

Preprocessing the input log information is essential to remove noise that can confuse the learning process and influence the results [28]. Experiments have shown that the learning result is sensitive to the input data, especially with noisy data. As shown in Table 2, there are many types of logs in the logging system of the routers and not all of them are conducive to the learning, especially when learning across a network with many routers. First, we only pick the types we want, which are Syslog, CPU utilization, LAN status and Memory utilization. During the learning phase, we can know when we do the exact actions and we can get the corresponding types of information or logs. With some test data that we generated by ourselves, the training dataset includes labeled data. We attacked the routers by using open source attack tools and our own codes. Thus, we could get the logs and data of the attack. For the benign action, we also obtained the logs from the admin point of view. However, we could not perform the learning with the raw log data because there are many redundant logs in the files. Thus, we needed to eliminate these useless copies. As there are also numerous variants in the syslog entries, we also eliminated the variants and extracted the log templates. We used regular expression to process the variants in the logs and extract the log templates from the raw log data.

### 4.2. Feature Vectorization

Table 3 shows the CPU utilization and the items in the table have different meanings. usr means CPU time spent in user space; sys means CPU time spent in kernel space; nice means CPU time spent on low priority processes; idle means CPU time spent idle; io means CPU time spent in IO waiting (on disk); hardirq means CPU time spent servicing/handling hardware interrupts; and softirq means CPU time spent servicing/handling software interrupts. In the table, there are seven items describing the utilization of CPU and we can form these items into a seven-dimension vector.

Table 4 is the memory allocation information. We can also form entries into vectors. The advantage of this method is that we do not need to have expert knowledge in advance and we can use its information vector to get the feature information relevant to the corresponding behavior.

For the Lan logs, we get some tuples from the log, such as TX Packet Sent, TX Underrun, TX Bus Error, RX Packet Received, RX Overflow, and RX Bus Error (TX, transport; RX, receive). Extracting the characteristics from syslogs is complicated because they are in the form of text. We performed this part of work in our former work Dlog [7]. There are many features we denoted for a log sequences, such as number of IP, time duration, severity level and so on. The feature vector is finally formed as a 17-dimension vector.

### 4.3. Clustering

Since we converted different types of log information into corresponding vectors, we should create different collections based on different events. However, the initial vector is not available for calculation. Because the dimensions in the vector are too high, it consumes a lot of computing resources and slows down the calculations. Therefore, we needed to convert high-dimensional vectors into low-dimensional vector datasets. It is often helpful to use a dimensionality-reduction technique such as PCA prior to performing machine learning [29]. PCA finds the principal components of the dataset and transforms the data into a new, lower-dimensional subspace, which will help improve the performance of clustering. The whole process is shown in Figure 2. As for the learning dataset, we learned 569 benign events and 154 anomalies from our own test codes and faults report. We could learn some known attacks and cluster them into categories before detection. However, for the novel malicious events, the approach based on history learning cannot gain a good result. Thus, we propose a cluster method based on log correlation. Traditional machine learning has difficulty identifying untrained data.

*A* is defined as an attack-related log set, and *B* is defined as a benign log set. In the optimal case, $|e_{A}|$≫$|e_{AB}|$ and $|e_{B}|$≫$|e_{AB}|$ ($|e_{A}|$ and $|e_{B}|$ represents the edge number among their own clusters and $|e_{AB}|$ represents the edge number between these two clusters). However, in real environment, the case $|e_{A}|$≪$|e_{AB}|$ often emerges and this will reduce the accuracy of identification of the clusters. To eliminate the above problem, we assigned corresponding weights to the edges and achieved the goal $\omega_{A}$·$|e_{A}|$ > $\omega_{AB}$·$|e_{AB}|$ ($\omega_{A}$ is the weight assigned to $|e_{A}|$ and $\omega_{AB}$ is the weight assigned to $|e_{AB}|$). That means the algorithm should assign a global weight vector $\vec{\alpha }$ to each of the edge. Thus, the equations still hold:
(1)$$\begin{array}{c} \sum_{e\epsilon e_{A}}^{}\sum_{i=1}^{k}\alpha_{i}\cdot e_{i}>\sum_{e\epsilon e_{AB}}^{}\sum_{i=1}^{k}\alpha_{i}\cdot e_{i}\\ \sum_{e\epsilon e_{B}}^{}\sum_{i=1}^{k}\alpha_{i}\cdot e_{i}>\sum_{e\epsilon e_{AB}}^{}\sum_{i=1}^{k}\alpha_{i}\cdot e_{i} \end{array}$$
where *k* is the dimension number of edge vector, $e_{i}$ is *i*th value of vector $\vec{e}$, and we set the dot product result as the weight of each edge (ω = $\sum_{i=1}^{k}\alpha_{i}\cdot e_{i}$). However, the weight can be negative value due to this method and ω is not the global optimal solution in graph. Thus, we converted the above formula into a quadratic optimal problem:
(2)$$\begin{array}{cc} max_{\vec{\alpha }}\sum_{e\epsilon e_{A}}^{}\sum_{i=1}^{k}\alpha_{i}\cdot e_{i}+\sum_{e\epsilon e_{AB}}^{}\sum_{i=1}^{k} & \alpha_{i}\cdot e_{i}-\lambda \sum_{e\epsilon e_{A}}^{}\sum_{i=1}^{k}\alpha_{i}\cdot e_{i}-\frac{1}{2}{\vec{\alpha }}^{T}\cdot \vec{\alpha }\\ & s.t.0\leq {\vec{\alpha }}^{T}e\leq 1 \end{array}$$
where λ is the trade-off parameter to balance between the first two terms and the third term in the target function. Regularizer $\frac{1}{2}{\vec{\alpha }}^{T}\cdot \vec{\alpha }$ is to avoid the over fitting problem. The target function is convex and the output weight vector $\vec{\alpha }$ is global optimum. After the graph is converted into a weight map, it is necessary to extract the cluster of points from the graph efficiently. This project uses the Louvain algorithm [30] for unsupervised node identification because the algorithm can efficiently process large-scale network nodes.

We define $A_{v,m}$ as the weight between node *v* and node *w*. $k_{v}$ = $\sum_{e\epsilon w}^{}A_{v,m}$ is the weight sum connected to node *w*. $c_{v}$ is the cluster where node *v* is assigned. Function δ means that if *i* = *j*, then δ(i,j) = 1, and δ(i,j) = 0 otherwise. Modularity is defined as:
(3)$$\begin{array}{c} Q=\frac{1}{2m}\sum_{v,m}^{}\left[A_{v,m}-\frac{k_{i}k_{j}}{2m}\right]\delta \left(c_{v},c_{w}\right),where,m\;\frac{1}{2m}\sum_{v,m}^{}A_{v,m} \end{array}$$

The modular value of the point ranges from −1 to 1, which is used to calculate the density value of the node’s connectivity in the graph. After the above formula initializes the module, the Louvain algorithm is used to perform the two-step operation of the greedy optimization: first, the point *v* is moved to the set *C* of its neighbor point *w*. Second, the maximum value of the modular change is evaluated by the algorithm, and  *v* moves to the set with the largest module change. The set obtained by the calculation is the respective classification set of the attack and the normal point.

### 4.4. Anomaly Detection

For each cluster, we calculate two centers: (1) K-means center $C_{mn}$: mean value of each point in the cluster; and (2) K-medoids $C_{md}$: a represented point in the cluster. Then, we calculate each node’s distances to both of the centers. We denote the point of a cluster(k) as $N_{x}^{k}$, which is shown in Equation (Equation 4):
(4)$$N_{x}^{k}\;\epsilon \;cluster\left(k\right),cluster\left(k\right)|=n,\;x=1,2,\cdots ,n\;$$

In cluster(k), the points have the same dimensions (*v*). We define the distance between two nodes $N_{i}^{k}$ = ($I_{i1}^{k}$, $I_{i2}^{k}$, ..., $I_{iv}^{k}$), $N_{j}^{k}$ = ($I_{j1}^{k}$, $I_{j2}^{k}$, ..., $I_{jv}^{k}$) as:
(5)$$d\left(N_{i}^{k},N_{j}^{k}\right)=\sqrt{{\left(I_{i1}^{k}-I_{j1}^{k}\right)}^{2}+{\left(I_{i2}^{k}-I_{j2}^{k}\right)}^{2}+\cdots +{\left(I_{iv}^{k}-I_{jv}^{k}\right)}^{2}}$$

As there are two centers, we calculate two radii of the cluster: one is to the K-means center $C_{mn}$ and the other is to the K-medoids $C_{md}$. Then, we compare these two radii and choose the maximal one as the threshold of the distance, which is shown in Equation (Equation 6).
(6)$$\begin{array}{c} T_{mn}^{k}=d\left(N_{I}^{k},C_{mn}^{k}\right),\;\mathrm{where}\;I=\arg\;\max\;d\left(N_{i}^{k},C_{mn}^{k}\right)\\ T_{md}^{k}=d\left(N_{I}^{k},C_{md}^{k}\right),\;\mathrm{where}\;I=\arg\;\max\;d\left(N_{i}^{k},C_{md}^{k}\right)\\ T_{h}=\mathrm{MAX}\left(T_{mn}^{k},T_{md}^{k}\right) \end{array}$$

As the input point $N_{x}$ arrives, we first cluster it to the nearest group and calculate the distance to the two centers. We choose the minimum one as the distance which is used for comparison with the threshold (Equation (Equation 7)). If the distance is larger than the threshold, then it cannot be a cluster in the normal event group and we can judge it as an abnormal event. Otherwise, it is the event that can be grouped into our learning one and it is normal, which is shown in Figure 3.
(7)$$\left\{\begin{array}{c} \mathrm{MIN}\left(d\left(N_{x},C_{mn}\right),d\left(N_{x},C_{md}\right)\right)>T_{h}:\mathrm{Abnormal}\\ \mathrm{MIN}\left(d\left(N_{x},C_{mn}\right),d\left(N_{x},C_{md}\right)\right)\leq T_{h}:\mathrm{Normal} \end{array}\right.$$

After we detected the attack, we matched it to the learned attack pattern. If the attack can match the template, we can tell the user which type the attack belongs to. Then, we can recommend the corresponding solution and patch for the attack. However, if the attack is not in the cluster set, we should send it to the appropriate expert for subsequent authentication. If the expert confirms the corresponding attack and can give a solution, we return this result to the training process, which can gradually improve the accuracy of the anomaly detection.

### 4.5. Attack Prediction

To predict the attack, we need to know the clues before the attack happens. Thus, we should find the relationships among the attack events and pre-attack events. We can refer to the records on router device. If an attack sniffs the routers, some types of log entries will record the clue on them (at least reflect the system conditions objectively) , but the syslogs or secure logs may not show these actions (they are printed according to the developers’ subjective intention). For example, if an adversary uses Nmap to scan the router, LAN status and CPU utilization of routers will turn to a different condition compared with the normal status. The primary solution is to detect this kind of regular pre-attack event. However, some steps before the attack are not malicious and they are just the normal process, but they are also the key steps of the attack chain. If we just detect the abnormalities, we will miss the whole attack chain. For example, after the attacker cracks the password of the router, he will also login into the system and this seems to be a normal action which none of the log entries will indicate as spurious. Constructing the whole logical attack chain and identifying the attacker’s action are challenges in our log correlation analysis work. In dynamic programming, we can find the longest common subsequence of two strings. For instance, $S_{1}$ is AqeTptQnF and $S_{2}$ is AxTbirQlcF, in which the LCS is ATQF. We can get the pre-attack condition sequence before the attack and each event or condition can be viewed as the letter in the above string. If we manage to find the same subsequence condition before one attack, we can identify the regular pre-steps before the attack. Primitively, we can leverage *divide and conquer* to get the common subsequence between two condition chains. If there are *N* chains in the attack sampling, the LCS calculation should be N−1. Although *divide and conquer* can theoretically outline the regular attack steps, the result may hide some different attack steps. For instance, an adversary intends to crack the login password of a router. He can take different ways to make it. In one way, he can exam the input parameters of the login page and do a brute force crack. In the other way, he can scan the open ports of the router. Then, he analyzes and discovers the vulnerabilities of the router. Finally, he launches the attack and manages to crack in by using the bugs of the system. Regular *divide and conquer* can eliminate pre-attack steps. Thus, we design the algorithm to determine the regular pre-attack steps.

With Algorithm 1, we can find the LCS of each of the two pre-attack’s history condition. The contents of the same attack (LS[*i*][]) may not have the common subsequences. Thus, we cannot do the LCS algorithm iteratively and we should construct the contents as a tree, which can show the attack path towards the attack.
**Algorithm 1** LCS cluster tree building algorithm.**Require** **:**Sampling cluster *A*, $C_{ij}$ϵ*A* (i=1 to *p* and j=1 to *x*)**Ensure** **:**The pre-attack steps array LS[][] of each attack $a_{i}$  1:**for** (i=1; i<p; *i*++) **do**  2: k=0;  3: **for** (m=2; m<=x; *m*++) **do**  4:  **for** (n=1; n<=m; *n*++) **do**  5:   *s* = $LCS(C_{im},C_{in})$;  6:   **if** (*s*!= null) **then**  7:   LS[*i*][*k*++]=*s*;  8:   Build_Tree(s);  9:**  end if** 10:**  end for** 11:** end for** 12:**end for**


## 5. Evaluation

We evaluated our approach in various routers including Huawei, Cisco, Tenda, TPlink and DLink. We first collected the logs of benign user actions and attacks to do the training step. Then, we calculated the distance between event and cluster to detect the anomalies. Finally, we used the routers conditions and information to find the regular pre-steps before the attack with the help longest common subsequences.

### 5.1. Experiment Setup

We set up the experimental platform as shown in Figure 4. We tried to reproduce the real adversary’s attack process. First, we collected the system information by active and passive information collection. We used some open source tools to collect the router’s information such as open port, system version, IP address, etc. With the system information, we mined the vulnerabilities of the system. Manually discovery and tool-support discovery were both leveraged in this step. By collecting the target system’s feedback, we could find some vulnerabilities of the system and prepare for the attack. During the attack, we used open source code, penetration tools or our own code to do the attack. We also obtained the feedback and system state of the router and evaluated the attack results. Finally, we also analyzed the most efficient attack methods and instructed the following adversary actions. We could construct a map of the attack and use the most efficient attack to destroy the router. Through this platform, we could get the corresponding logs and system data of the attack and malicious behavior. We configured the syslog options of the routers in the network and made them send the logs to our syslog server, and we used the SQL server to record the log files. As shown in Table 5, we used different tools or methods to launch a variety of attacks towards the routers. We tried to crack the routers’ login user name and password by our own java codes using the format of the data post to the router’s web front. The other attacks were launched by the tools in the platform of Windows 7 or Ubuntu 12.07.

### 5.2. Performance of Multiple Information Learning

During the learning phase, we evaluated the turnaround time and memory costs of data cleaning, feature vectorization and clustering. As shown in Figure 5, we selected the logs which range from 10,000 to 90,000 and calculated the corresponding costs. We compared the time cost with Dlog and we could achieve apparent lower time cost. In the data cleaning, we eliminated the redundant log copies and processed the variants in the logs according to the regular expressions. With regular expressions, we could obviously reduce the time of log templates extraction. The time cost of feature vectorization occupied the most turn-around time of the learning. We should process the CPU utilization, LAN status, Memory utilization and syslog, which would cost a lot of time and memory space. We used DBSCAN as the cluster algorithm and found that the time cost rose quickly as the log increased. The time complexity of DBSCAN is $O(n^{2})$, as shown in Figure 5.

### 5.3. Time Window Setting

It is important to set the window size during the identification of the events from logs. If we set the size of window too small, the events would be break up and this would influence the accuracy of events detection. Otherwise, if we set the window too big, there would be more than one events in one log chunk and we would not be able to identify the real events from the log. Thus, deciding the time window size is critical in identifying the events. We arranged the window size from 0 to 120 s and obtained the experiment results in Figure 6. We achieved relatively high detection accuracy between the window size 35 s and 50 s when we used the raw log size of 10,000. The duration of the average event was 1 min. Thus, we could get the accuracy rate from 96.5% to 80%.

### 5.4. Anomaly Detection Accuracy

We also evaluated the performance of anomaly detection of our approach. A DOS attack was launched towards routers and we leveraged different approaches to the learning and diagnosis steps. Then, we gathered the key-values of recall and precision of the attack detection, which were calculated as Equations (Equation 8) and (Equation 9), and compared their results.
(8)$$Precision=\frac{Correctly\;detected\;attacks}{Total\;inserted\;attacks}$$
(9)$$Recall=\frac{Correctly\;detected\;attacks}{Total\;detected\;attacks}$$

We aligned our former work SyslogDigest [13] and NetPro [31] as the baselines. NetPro conducts confidential inferencing and verification by using the pre-defined communication rules. It can find the positive and passive attacks of the routers in the MANETs. The rules are denoted according to the communication rules by the expert. To start the reasoning process, the raw unstructured logs must be transformed to high-level structured rules. In this process, there is time cost in the transformation and it is easy to make some faults. SyslogDigest extracts the log templates and detects the network events with the sub-type tree of the log sequences. To use the method of SyslogDigest, the primary requirement is that the messages must be the same log type. However, the log type cannot be decided in advance. Thus, it is impractical to cluster them in the same structure.

Figure 7 shows that our approach can improve the anomaly detection and the average accuracy rate was 89.6% which is 5.1% higher than NetPro. Compared with SyslogDigest, our approach had a lower false positive rate.

### 5.5. Attack Prediction Performance

We performed the experiment to show the performance of attack prediction, as shown in Figure 8. We launched these seven attacks during three months and we learned the corresponding characteristics of these attacks. We chose the points ranging from 0.1 to 1 in recall and counted the average values. We could gain relatively good feedback in predicting IP spoofing attack and ARP spoofing attack, as shown in Figure 8. We analyzes the form of these logs and found that the structure of the log chunks are so unique when the routers are under these two attacks. Thus, we could identify the pre-attack steps and predict them precisely. Predicting password violence crack login and WiFi had lower precision and recall than the former two attacks because sometimes our approach regarded the normal login faults or device connection as the attack. We also noticed that predicting the DOS attack was not as good as in ARP spoofing attack. As we referred to the logs of the DOS attack, we found that the log chunks are sometimes the subset of the SSL spoofing attack. Thus, our approach would predict it falsely. In our future work, we should mitigate the above faults and improve our detection and prediction approach to satisfy more rigorous conditions. As we accumulate more training datasets, deep learning should be considered and we can also combine the idea of blockchain to conduct the data diagnosis work.

## 6. Conclusions

To detect the anomalies and predict an attack on the routers, we propose an approach using multi-source syslogs and log correlation. The approach in this paper consists of three main steps. First, we collect and formalize the data and we do the training step. We extract the characteristics and form the log sequences as vectors. Second, we calculate the events’ distance from the cluster and decide if it is an anomaly. Third, we innovatively transform the attack chain construction problem as the longest common subsequences finding problem. We analyze the syslogs according to the conditions and extract the log sequences from the event. Our approach analyzes different types of logs from multiple angles and granularities. With the attack chain, we can protect the routers before the attack happens. As our experimental results show, our approach could improve the anomaly detection accuracy rate by about 5.1% compared to the former work and it is practical to be used in a real router environment.

## Figures and Tables

**Figure 1 entropy-21-00734-f001:**
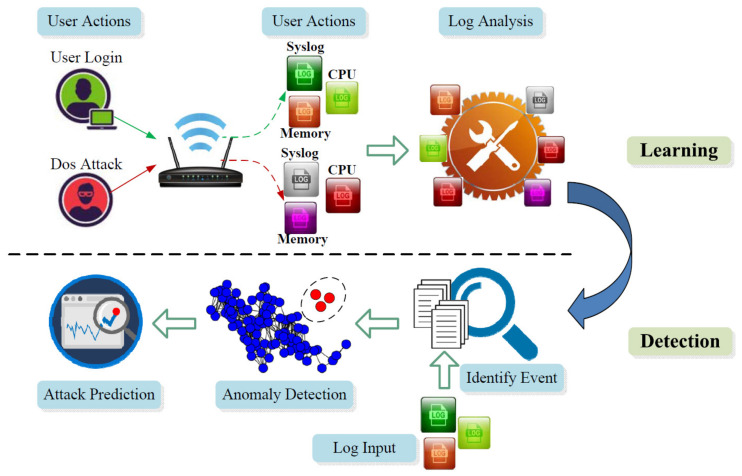
Work flow of learning and detection.

**Figure 2 entropy-21-00734-f002:**
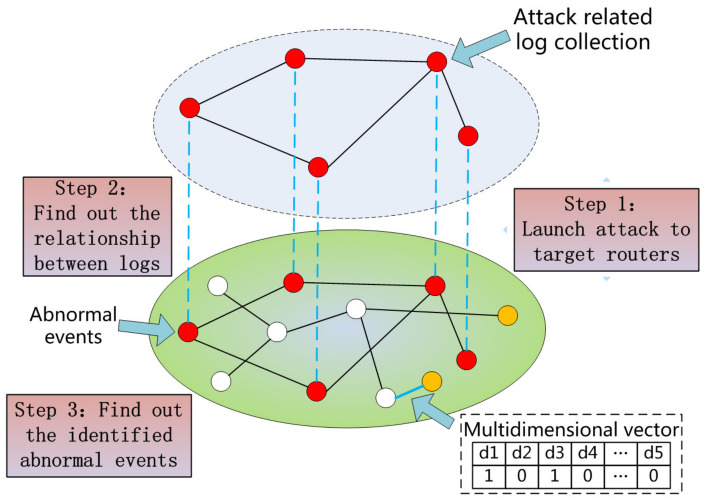
Log collaborative association analysis.

**Figure 3 entropy-21-00734-f003:**
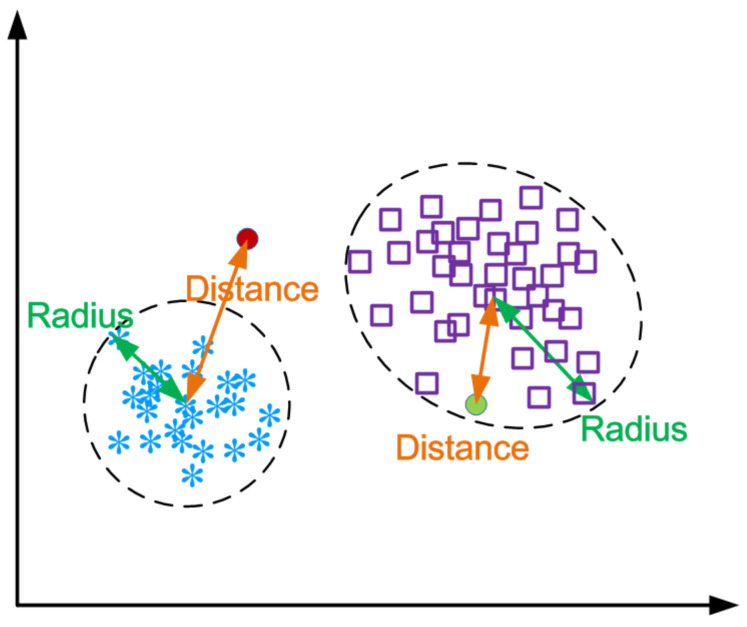
Anomaly detection based on the distance (example of two dimensions).

**Figure 4 entropy-21-00734-f004:**
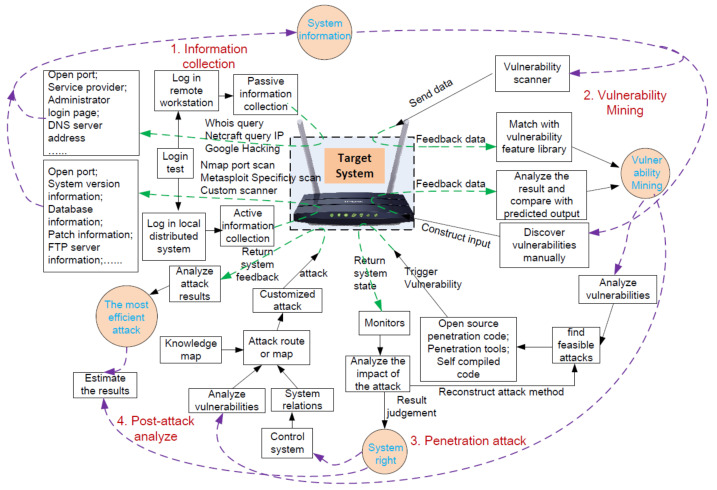
Work flow of the attack towards routers.

**Figure 5 entropy-21-00734-f005:**
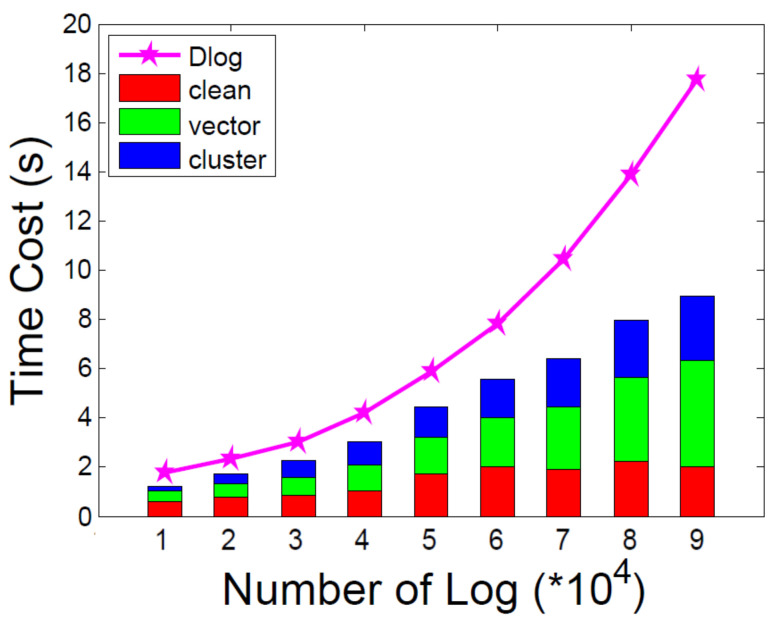
Time cost of multiple information learning.

**Figure 6 entropy-21-00734-f006:**
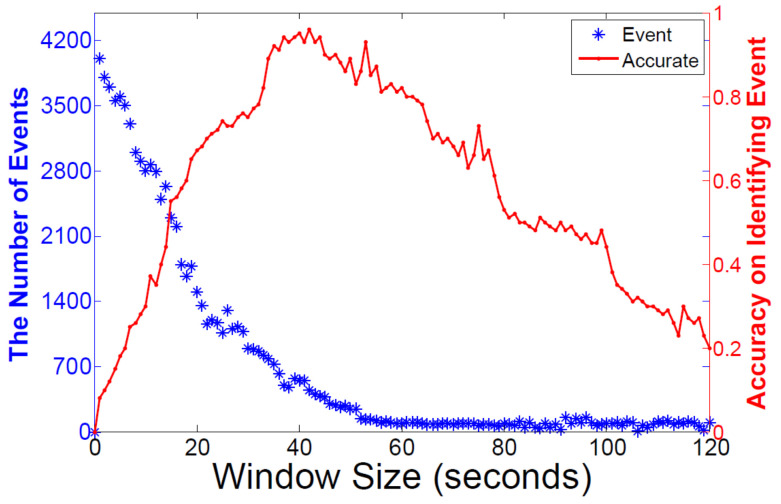
Time window influences on identifying event number and accuracy.

**Figure 7 entropy-21-00734-f007:**
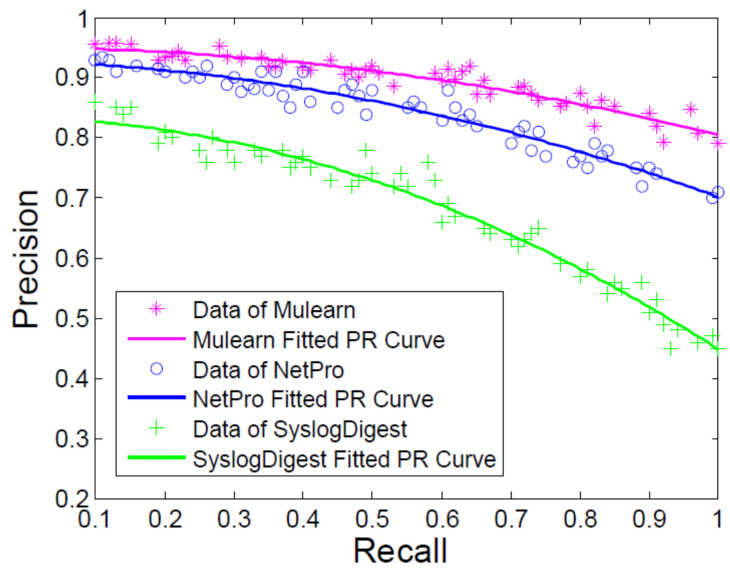
Precision and recall of attack detection.

**Figure 8 entropy-21-00734-f008:**
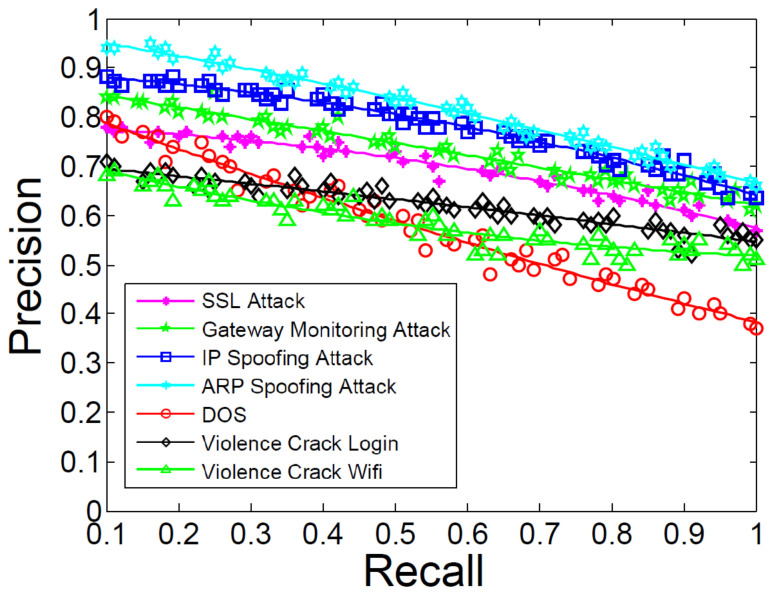
Precision and recall of attack clustering.

**Table 1 entropy-21-00734-t001:** Different router attack events.

Brand	Attack Event	Router Type	Time	Number of Influence
DLink	DNS Spoofing	DSL-2640B	April 2019	15,000
HuaWei	Botnet	HG523a	March 2019	18,000
Cisco	Remote Command Execution	RV320, RV325	January 2019	10,000
TP-LINK	Unrestricted remote control	TL-WR940N	February 2019	15,000
Netgear	Auth Bypass and Loads of Other Bugs	RT1900ac	February 2018	12,000

**Table 2 entropy-21-00734-t002:** Log correlation table.

Action	Firewall	DNS	CPU Utilization	FTP	LAN Status	Auth	Memory
Admin Login			x		x	x	
Wireless Connect			x		x		x
Remote Control		x		x	x		x
DOS Attack	x		x		x		x
Violence Crack Login			x		x	x	x
ARP Spoofing	x	x	x		x		x

**Table 3 entropy-21-00734-t003:** Examples of CPU  utilization.

Time	%usr	%sys	%nice	%idle	%io	%hardirq	%softirq
19:45:18	10.4%	85.1%	0%	5.8%	3.3%	1.2%	0%
19:45:20	10%	85.7%	0%	4.5%	3%	1.3%	0%
19:45:22	9.7%	86%	0%	6.1%	2.7%	1.6%	0%

**Table 4 entropy-21-00734-t004:** Examples of memory utilization.

Time	Used	Free	Shared	Buff	Cached
19:45:18	85,396 K	34,592 K	0 K	9216 K	23,888 K
19:45:20	95,264 K	19,820 K	0 K	8596 K	21,920 K
19:45:22	95,432 K	19,652 K	0 K	8596 K	22,076 K

**Table 5 entropy-21-00734-t005:** Attacks and tools.

No.	Attack Types	Method or Tool	OS
1	Login Password Violence Crack	Java code	Windows 7
2	Wifi Password Violence Crack	fern-wifi-cracker	Ubuntu 12.07
3	IP Spoofing Attack	Nmap	Windows 7
4	SSL Attack	THC-SSL	Windows 7
5	DOS Attack	HULK	Ubuntu 12.07
6	ARP Spoofing Attack	WinArpAttacker	Windows 7
7	Gateway Monitoring Attack	WinArpAttacker	Windows 7

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
