# Peer review of "Anomalies Detection and Proactive Defence of Routers Based on Multiple Information Learning†"

_entropy, 2019, doi:10.3390/e21080734_

Round 1
Reviewer 1 Report
Summary. The manuscript presents a model to detect anomalies detection in a network environment. The idea is to utilize multiple sources of data and more specifically log files to train a detection model and thus be able to detect future attacks before their occurrences.
Evaluation. The idea presented in this paper (i.e., learning from multiple sources) is not novel. The exact machine learning algorithms used in this paper is not clear either. It seems the idea discussed in this paper is not exactly a machine learning approach. Rather it is a data mining approach based on cluster analysis and more specifically based on DBSCAN algorithm. In simple words, it seems the entire idea is to capture log files from various sources, cluster them and then use the clustering to predict the future events in terms of being malicious attacks or benign. Along this line, there is not much novelty presented in this paper and it is just reporting the experiments conducted in this paper.
There are several other concerns, as follows:
1) it is not clear whether the training datasets are labeled or not.
2) It is not clear how different types of attacks are generated by admin.
3) It is not clear how “balanced” datasets (equal number of attacks and normal datasets) are generated.
4) The use of “control flow graph and analysis” is inappropriate in this context. Control flow is for the control of execution of a given program. However, in here the use of log data looks more like “data flow analysis” than execution. Therefore, it seems data flow analysis should be used.
5) The creation and use of Table 3 is not clear.
6) Table 4 has 5 columns, the description has 7 columns.
7) How Table 4 and 5 have been produced?
8) In formula 2, what are those I_{i}s?
9) Figure 6 needs explanation.
10) Section 4.4, formulas 5 – 6 are not used.,
Reviewer 2 Report
The english of the paper is often very hard to read and to understand, in a couple of points you seems to write the opposite of what you mean.
Round 2
Reviewer 2 Report
The new version is much more clear and easy to understand.
Congratulations for the hard work.
